# Finding a sparse vector in a subspace:
# Linear sparsity using alternating directions

**Qing Qu, Ju Sun, and John Wright**
{qq2105, js4038, jw2966}@columbia.edu
Dept. of Electrical Engineering, Columbia University, New York City, NY, USA, 10027

## Abstract

We consider the problem of recovering the sparsest vector in a subspace $\mathcal{S} \in \mathbb{R}^p$ with $\dim(\mathcal{S}) = n$. This problem can be considered a homogeneous variant of the sparse recovery problem, and finds applications in sparse dictionary learning, sparse PCA, and other problems in signal processing and machine learning. Simple convex heuristics for this problem provably break down when the fraction of nonzero entries in the target sparse vector substantially exceeds $1/\sqrt{n}$. In contrast, we exhibit a relatively simple nonconvex approach based on alternating directions, which provably succeeds even when the fraction of nonzero entries is $\Omega(1)$. To our knowledge, this is the first practical algorithm to achieve this linear scaling. This result assumes a planted sparse model, in which the target sparse vector is embedded in an otherwise random subspace. Empirically, our proposed algorithm also succeeds in more challenging data models arising, e.g., from sparse dictionary learning.

## 1 Introduction

Suppose we are given a linear subspace $\mathcal{S}$ of a high-dimensional space $\mathbb{R}^p$, which contains a sparse vector $\mathbf{x}_0 \neq \mathbf{0}$. Given arbitrary basis of $\mathcal{S}$, can we efficiently recover $\mathbf{x}_0$? Equivalently, provided a matrix $\mathbf{A} \in \mathbb{R}^{(p-n) \times p}$, can we efficiently find a nonzero sparse vector $\mathbf{x}$ such that $\mathbf{A}\mathbf{x} = \mathbf{0}$? In the language of sparse approximation, can we solve

$$\min_{\mathbf{x}} \ \|\mathbf{x}\|_0 \quad \text{s.t.} \quad \mathbf{A}\mathbf{x} = \mathbf{0}, \ \mathbf{x} \neq \mathbf{0} \qquad ? \tag{1}$$

Variants of this problem have been studied in the context of applications to numerical linear algebra [15], graphical model learning [27], nonrigid structure from motion [16], spectral estimation and Prony's problem [11], sparse PCA [29], blind source separation [28], dictionary learning [24], graphical model learning [3], and sparse coding on manifolds [21].

However, in contrast to the standard sparse regression problem ($\mathbf{A}\mathbf{x} = \mathbf{b}, \mathbf{b} \neq \mathbf{0}$), for which convex relaxations perform nearly optimally for broad classes of designs $\mathbf{A}$ [14, 18], the computational properties of problem (1) are not nearly as well understood. It has been known for several decades that the basic formulation

$$\min_{\mathbf{x}} \|\mathbf{x}\|_0, \quad \text{s.t.} \quad \mathbf{x} \in \mathcal{S} \setminus \{\mathbf{0}\}, \tag{2}$$

is NP-hard [15]. However, it is only recently that efficient computational surrogates with nontrivial recovery guarantees have been discovered. In the context of sparse dictionary learning, Spielman et al. [24] introduced a relaxation which replaces the nonconvex problem (2) with a sequence of linear programs:

$$\min_{\mathbf{x}} \|\mathbf{x}\|_1, \quad \text{s.t.} \quad x_i = 1, \ \mathbf{x} \in \mathcal{S}, \ 1 \leq i \leq p, \tag{3}$$

and proved that when $\mathcal{S}$ is generated as a span of $n$ random sparse vectors, with high probability the relaxation recovers these vectors, provided the probability of an entry being nonzero is at most $\theta \in O(1/\sqrt{n})$.

In a *planted sparse model*, in which $\mathcal{S}$ consists of a single sparse vector $\mathbf{x}_0$ embedded in a "generic" subspace, Hand et al. proved that (3) also correctly recovers $\mathbf{x}_0$, provided the fraction of nonzeros in $\mathbf{x}_0$ scales as $\theta \in O\left(1/\sqrt{n}\right)$ [19].

Unfortunately, the results of [24, 19] are essentially sharp: *when $\theta$ substantially exceeds $1/\sqrt{n}$, in both models the relaxation* (3) *provably breaks down.* Moreover, the most natural semidefinite programming relaxation of (1),

$$\min_{\mathbf{X}} \|\mathbf{X}\|_1, \quad \text{s.t.} \quad \left\langle \mathbf{A}^\top \mathbf{A}, \mathbf{X} \right\rangle = 0, \ \text{trace}[\mathbf{X}] = 1, \ \mathbf{X} \succeq \mathbf{0}. \tag{4}$$

also breaks down at exactly the same threshold of $\theta \sim 1/\sqrt{n}$.[1]

One might naturally conjecture that this $1/\sqrt{n}$ threshold is simply an intrinsic price we must pay for having an efficient algorithm, even in these random models. Some evidence towards this conjecture might be borrowed from the surface similarity of (2)-(4) and *sparse PCA* [29]. In sparse PCA, there is a substantial gap between what can be achieved with efficient algorithms and the information theoretic optimum [10]. Is this also the case for recovering a sparse vector in a subspace? *Is $\theta \in O\left(1/\sqrt{n}\right)$ simply the best we can do with efficient, guaranteed algorithms?*

Remarkably, this is not the case. Recently, Barak et al. introduced a new rounding technique for sum-of-squares relaxations, and showed that the sparse vector $\mathbf{x}_0$ in the planted sparse model can be recovered when $p \geq \Omega\left(n^2\right)$ and $\theta \geq \Omega(1)$ [8]. It is perhaps surprising that this is possible at all with a polynomial time algorithm. Unfortunately, the runtime of this approach is a high-degree polynomial in $p$, and so for machine learning problems in which $p$ is either a feature dimension or sample size, this algorithm is of theoretical interest only. However, it raises an interesting algorithmic question: *Is there a practical algorithm that provably recovers a sparse vector with $\theta \gg 1/\sqrt{n}$ nonzeros from a generic subspace $\mathcal{S}$?*

In this paper, we address this problem, under the following hypotheses: we assume the planted sparse model, in which a target sparse vector $\mathbf{x}_0$ is embedded in an otherwise random $n$-dimensional subspace of $\mathbb{R}^p$. We allow $\mathbf{x}_0$ to have up to $\theta_0 p$ nonzero entries, where $\theta_0$ is a constant. We provide a relatively simple algorithm which, with very high probability, exactly recovers $\mathbf{x}_0$, provided that $p \geq \Omega\left(n^4 \log^2 n\right)$.

Our algorithm is based on alternating directions, with two special twists. First, we introduce a special data driven initialization, which seems to be important for achieving $\theta = \Omega(1)$. Second, our theoretical results require a second, linear programming based rounding phase, which is similar to [24]. Our core algorithm has very simple iterations, of linear complexity in the size of the data, and hence should be scalable to moderate-to-large scale problems.

In addition to enjoying theoretical guarantees in a regime ($\theta = \Omega(1)$) that is out of the reach of previous practical algorithms, it performs well in simulations – succeeding empirically with $p \geq \Omega\left(n \log n\right)$. It also performs well empirically on more challenging data models, such as the dictionary learning model, in which the subspace of interest contains not one, but $n$ target sparse vectors. Breaking the $O(1/\sqrt{n})$ sparsity barrier with a practical algorithm is an important open problem in the nascent literature on algorithmic guarantees for dictionary learning [5, 4, 2, 1]. We are optimistic that the techniques introduced here will be applicable in this direction.

## 2 Problem Formulation and Global Optimality

We study the problem of recovering a sparse vector $\mathbf{x}_0 \neq \mathbf{0}$ (up to scale), which is an element of a known subspace $\mathcal{S} \subset \mathbb{R}^p$ of dimension $n$, provided an arbitrary orthonormal basis $\mathbf{Y} \in \mathbb{R}^{p \times n}$ for $\mathcal{S}$. Our starting point is the nonconvex formulation (2). Both the objective and constraint are nonconvex, and hence not easy to optimize over. We relax (2) by replacing the $\ell^0$ norm with the $\ell^1$ norm. For the constraint $\mathbf{x} \neq \mathbf{0}$, which is necessary to avoid a trivial solution, we force $\mathbf{x}$ to live on the unit sphere $\|\mathbf{x}\|_2 = 1$, giving

$$\min_{\mathbf{x}} \ \|\mathbf{x}\|_1, \quad \text{s.t.} \quad \mathbf{x} \in \mathcal{S}, \ \|\mathbf{x}\|_2 = 1. \tag{5}$$

This formulation is still nonconvex, and so we should not expect to obtain an efficient algorithm that can solve it globally for general inputs $\mathcal{S}$. Nevertheless, the geometry of the sphere is benign enough that for well-structured inputs it actually *will* be possible to give algorithms that find the global optimum of this problem.

The formulation (5) can be contrasted with (3), in which we optimize the $\ell^1$ norm subject to the constraint $\|\mathbf{x}\|_\infty = 1$. Because $\|\cdot\|_\infty$ is polyhedral, that formulation immediately yields a sequence of linear programs. This is very convenient for computation and analysis, but suffers from the aforementioned breakdown behavior around $\|\mathbf{x}_0\|_0 \sim p/\sqrt{n}$.

In contrast, the sphere $\|\mathbf{x}\|_2 = 1$ is a more complicated geometric constraint, but will allow much larger numbers of nonzeros in $\mathbf{x}_0$. For example, if we consider the global optimizer of a variant of (5):

$$\min_{\mathbf{q}\in\mathbb{R}^n} \|\mathbf{Y}\mathbf{q}\|_1, \quad \text{s.t.} \quad \|\mathbf{q}\|_2 = 1, \tag{6}$$

under the *planted sparse model* (detailed below), $\mathbf{e}_1$ is the unique to (6) with very high probability:

**Theorem 2.1** ($\ell^1/\ell^2$ recovery, planted sparse model). *There exists a constant $\theta_0 \in (1/\sqrt{n}, 1/2)$ such that if the subspace $\mathcal{S}$ follows the planted sparse model*

$$\mathcal{S} = \text{span}\,(\mathbf{x}_0, \mathbf{g}_1, \ldots, \mathbf{g}_{n-1}) \subset \mathbb{R}^p, \tag{7}$$

*with $\mathbf{g}_i \sim_{i.i.d.} \mathcal{N}(0, 1/p)$, and $\mathbf{x}_0 \sim_{i.i.d.} \frac{1}{\sqrt{\theta p}}\text{Ber}(\theta)$, with $\mathbf{x}_0, \mathbf{g}_1, \ldots, \mathbf{g}_{n-1}$ mutually independent and $1/\sqrt{n} < \theta < \theta_0$, then $\pm\mathbf{e}_0$ are the only global minimizers to (6) if $\mathbf{Y} = [\mathbf{x}_0, \mathbf{g}_1, \ldots, \mathbf{g}_{n-1}]$, provided $p \geq \Omega\,(n \log n)$.*

Hence, *if* we could find the global optimizer of (6), we would be able to recover $\mathbf{x}_0$ whose number of nonzero entries is quite large – even linear in the dimension $p$ ($\theta = \Omega(1)$). On the other hand, it is not obvious that this should be possible: (6) is nonconvex. In the next section, we will describe a simple heuristic algorithm for (a near approximation of) the $\ell^1/\ell^2$ problem (6), which guarantees to find a stationary point. More surprisingly, we will then prove that for a class of random problem instances, this algorithm, plus an auxiliary rounding technique, actually recovers the global optimum – the target sparse vector $\mathbf{x}_0$. The proof requires a detailed probabilistic analysis, which is sketched in Section 4.2.

Before continuing, it is worth noting that the formulation (5) is in no way novel – see, e.g., the work of [28] in blind source separation for precedent. However, the novelty originates from our algorithms and subsequent analysis.

## 3 Algorithm based on Alternating Direction Method (ADM)

To develop an algorithm for solving (6), we work with the orthonormal basis $\mathbf{Y} \in \mathbb{R}^{p\times n}$ for $\mathcal{S}$. For numerical purposes, and also for coping with noise in practical application, it is useful to consider a slight relaxation of (6), in which we introduce an auxiliary variable $\mathbf{x} \approx \mathbf{Y}\mathbf{q}$:

$$\min_{\mathbf{q},\mathbf{x}} \frac{1}{2} \|\mathbf{Y}\mathbf{q} - \mathbf{x}\|_2^2 + \lambda \|\mathbf{x}\|_1, \quad \text{s.t.} \quad \|\mathbf{q}\|_2 = 1, \tag{8}$$

Here, $\lambda > 0$ is a penalty parameter. It is not difficult to see that this problem is equivalent to minimizing the *Huber $m$-estimator* over $\mathbf{Y}\mathbf{q}$. This relaxation makes it possible to apply alternating direction method to this problem, which, starting from some initial point $\mathbf{q}^{(0)}$, alternates between optimizing with respect to $\mathbf{x}$ and optimizing with respect to $\mathbf{q}$:

$$\mathbf{x}^{(k+1)} = \arg\min_{\mathbf{x}} \frac{1}{2} \left\|\mathbf{Y}\mathbf{q}^{(k)} - \mathbf{x}\right\|_2^2 + \lambda \|\mathbf{x}\|_1, \tag{9}$$

$$\mathbf{q}^{(k+1)} = \arg\min_{\mathbf{q}} \frac{1}{2} \left\|\mathbf{Y}\mathbf{q} - \mathbf{x}^{(k+1)}\right\|_2^2 \text{ s.t. } \|\mathbf{q}\|_2 = 1. \tag{10}$$

Both (9) and (10) have simple closed form solutions:

$$\mathbf{x}^{(k+1)} = S_\lambda[\mathbf{Y}\mathbf{q}^{(k)}], \qquad \mathbf{q}^{(k+1)} = \frac{\mathbf{Y}^\top \mathbf{x}^{(k+1)}}{\left\|\mathbf{Y}^\top \mathbf{x}^{(k+1)}\right\|_2}, \tag{11}$$

---

**Algorithm 1** Nonconvex ADM

---

**Input:**    A matrix $\mathbf{Y} \in \mathbb{R}^{p \times n}$ with $\mathbf{Y}^\top \mathbf{Y} = \mathbf{I}$, initialization $\mathbf{q}^{(0)}$, threshold $\lambda > 0$.
**Output:**    The recovered sparse vector $\hat{\mathbf{x}}_0 = \mathbf{Y}\mathbf{q}^{(k)}$
 1: Set $k = 0$,
 2: **while** not converged **do**
 3:     $\mathbf{x}^{(k+1)} = S_\lambda[\mathbf{Y}\mathbf{q}^{(k)}]$,
 4:     $\mathbf{q}^{(k+1)} = \dfrac{\mathbf{Y}^\top \mathbf{x}^{(k+1)}}{\left\|\mathbf{Y}^\top \mathbf{x}^{(k+1)}\right\|_2}$,
 5:     Set $k = k + 1$.
 6: **end while**

---

where $S_\lambda[x] = \operatorname{sign}(x) \max\{|x| - \lambda, 0\}$ is the soft-thresholding operator. The proposed ADM algorithm is summarized in Algorithm 1.

For general input $\mathbf{Y}$ and initialization $\mathbf{q}^{(0)}$, Algorithm 1 is guaranteed to produce a stationary point of problem (8). This is a consequence of recent general analyses of alternating direction methods for nonsmooth and nonconvex problems – see [6, 7]. However, if our goal is to recover the *sparsest* vector $\mathbf{x}_0$, some additional tricks are needed.

**Initialization.**    Because the problem (6) is nonconvex, an arbitrary or random initialization is unlikely to produce a global minimizer.[2] Therefore, good initializations are critical for the proposed ADM algorithm to succeed. For this purpose, we suggest to use every normalized row of $\mathbf{Y}$ as initializations for $\mathbf{q}$, and solve a sequence of $p$ nonconvex programs (6) by the ADM algorithm.

To get an intuition of why our initialization works, recall the planted sparse model: $\mathcal{S} = \operatorname{span}(\mathbf{x}_0, \mathbf{g}_1, \ldots, \mathbf{g}_{n-1})$. Write $\mathbf{Z} = [\mathbf{x}_0 \mid \mathbf{g}_1 \mid \cdots \mid \mathbf{g}_{n-1}] \in \mathbb{R}^{p \times n}$. Suppose we take a row $\mathbf{z}_i$ of $\mathbf{Z}$, in which $\mathbf{x}_0(i)$ is nonzero, then $\mathbf{x}_0(i) = \Theta\left(1/\sqrt{\theta p}\right)$. Meanwhile, the entries of $\mathbf{g}_1(i), \ldots \mathbf{g}_{n-1}(i)$ are all $\mathcal{N}(0, 1/p)$, and so have size about $1/\sqrt{p}$. Hence, when $\theta$ is not too large, $\mathbf{x}_0(i)$ will be somewhat bigger than most of the other entries in $\mathbf{z}_i$. Put another way, $\mathbf{z}_i$ *is biased towards the first standard basis vector* $\mathbf{e}_1$.

Now, under our probabilistic assumptions, $\mathbf{Z}$ is very well conditioned: $\mathbf{Z}^\top \mathbf{Z} \approx \mathbf{I}$.[3] Using, e.g., Gram-Schmidt, we can find a basis $\bar{\mathbf{Y}}$ for $\mathcal{S}$ of the form

$$\bar{\mathbf{Y}} = \mathbf{Z}\mathbf{R}, \tag{12}$$

where $\mathbf{R}$ is upper triangular, and $\mathbf{R}$ is itself well-conditioned: $\mathbf{R} \approx \mathbf{I}$. Since the $i$-th row of $\mathbf{Z}$ is biased in the direction of $\mathbf{e}_1$ and $\mathbf{R}$ is well-conditioned, the $i$-th row $\bar{\mathbf{y}}_i$ is also biased in the direction of $\mathbf{e}_1$.

We know that the global optimizer $\mathbf{q}_\star$ should satisfy $\bar{\mathbf{Y}}\mathbf{q}_\star = \mathbf{x}_0$. Since $\mathbf{Z}\mathbf{e}_1 = \mathbf{x}_0$, we have $\mathbf{q}_\star = \mathbf{R}^{-1}\mathbf{e}_1 \approx \mathbf{e}_1$. Here, the approximation comes from $\mathbf{R} \approx \mathbf{I}$. Hence, for this particular choice of $\mathbf{Y}$, described in (12), *the $i$-th row is biased in the direction of the global optimizer*. This is what makes the rows of $\mathbf{Y}$ a particularly effective choice for initialization.

What if we are handed some other basis $\mathbf{Y} = \bar{\mathbf{Y}}\mathbf{U}$, where $\mathbf{U}$ is an orthogonal matrix? Suppose $\mathbf{q}_\star$ is a global optimizer to (6) with input matrix $\bar{\mathbf{Y}}$, then it is easy to check that, with input matrix $\mathbf{Y}$, $\mathbf{U}^\top \mathbf{q}_\star$ is also a global optimizer to (6), which implies that our initialization is *invariant* to any rotation of the basis. Hence, *even if we are handed an arbitrary basis for $\mathcal{S}$, the $i$-th row is still biased in the direction of the global optimizer*.

**Rounding.**    Let $\bar{\mathbf{q}}$ denote the output of Algorithm 1. We will prove that with our particular initialization and an appropriate choice of $\lambda$, the solution of our ADM algorithm falls within a certain radius of the globally optimal solution $\mathbf{q}_\star$ to (6). To recover $\mathbf{q}_\star$, or equivalently to recover the sparse vector $\mathbf{x}_0 = \mathbf{Y}\mathbf{q}_\star$, we solve the linear program

$$\min_{\mathbf{q}} \|\mathbf{Y}\mathbf{q}\|_1 \quad \text{s.t.} \quad \langle \mathbf{r}, \mathbf{q} \rangle = 1, \tag{13}$$

with $\mathbf{r} = \bar{\mathbf{q}}$. We will prove that if $\mathbf{r}$ is close enough to $\mathbf{q}_\star$, then this relaxation exactly recovers $\mathbf{q}_\star$, and hence $\mathbf{x}_0$.

# 4 Analysis

## 4.1 Main Results

In this section, we describe our main theoretical result, which shows that with high probability, the algorithm described in the previous section succeeds.

**Theorem 4.1.** *Suppose that $\mathcal{S}$ satisfies the planted sparse model, and let $\mathbf{Y}$ be an arbitrary basis for $\mathbf{S}$. Let $\mathbf{y}_1 \ldots \mathbf{y}_p \in \mathbb{R}^n$ denote the (transposes of) the rows of $\mathbf{Y}$. Apply Algorithm 1 with $\lambda = 1/\sqrt{p}$, using initializations $\mathbf{q}^{(0)} = \mathbf{y}_1, \ldots, \mathbf{y}_p$, to produce outputs $\bar{\mathbf{q}}_1, \ldots, \bar{\mathbf{q}}_p$. Solve the linear program (13) with $\mathbf{r} = \bar{\mathbf{q}}_1, \ldots, \bar{\mathbf{q}}_p$, to produce $\hat{\mathbf{q}}_1, \ldots, \hat{\mathbf{q}}_p$. Set $i^\star \in \arg\min_i \|\mathbf{Y}\hat{\mathbf{q}}_i\|_0$. Then*

$$\mathbf{Y}\hat{\mathbf{q}}_{i^\star} = \gamma \mathbf{x}_0, \tag{14}$$

*for some $\gamma \neq 0$, with overwhelming probability, provided*

$$p > Cn^4 \log^2 n, \qquad and \qquad \frac{1}{4\sqrt{n}} \leq \theta \leq \theta_0. \tag{15}$$

*Here, $C$ and $\theta_0 > 0$ are universal constants.*

We can see that the result in Theorem 4.1 is suboptimal compared to the global optimality condition and Barak et al.'s result in the sense of the sampling complexity that we require $p \geq Cn^4 \log^2 n$. While for the global optimality condition, we only need $p > Cn$ to guarantee a global optimal solution exists with high probability. For Barak et al.'s result, we need $p > Cn^2$. Nonetheless, compared to Barak et al., we believe this is the first practical and efficient method that is guaranteed to achieve $\theta \sim O(1)$ rate. The lower bound on $\theta$ in Theorem 4.1 is mostly for convenience in the proof; in fact, the LP rounding stage of our algorithm already succeeds with high probability when $\theta \in O(1/\sqrt{n})$.

## 4.2 A Sketch of Analysis

The proof of our main result requires rather detailed technical analysis of the iteration-by-iteration properties of Algorithm 1. In this subsection, we briefly sketch the main ideas. For detailed proofs, please see the technical supplement to this paper.

As noted in Section 3, the ADM algorithm is invariant to change of basis. So, we can assume without loss of generality that we are working with the particular basis $\bar{\mathbf{Y}} = \mathbf{ZR}$ defined in that section. In order to further streamline the presentation, we are going to sketch the proof under the assumption that

$$\mathbf{Y} = [\mathbf{x}_0 \mid \mathbf{g}_1 \mid \cdots \mid \mathbf{g}_{n-1}], \tag{16}$$

rather than the orthogonalized version $\bar{\mathbf{Y}}$. This may seem plausible, but when $p$ is large $\mathbf{Y}$ is already nearly orthogonal, and hence $\mathbf{Y}$ is very close to $\bar{\mathbf{Y}}$. In fact, in our proof, we simply carry through the argument for $\mathbf{Y}$, and then note that $\mathbf{Y}$ and $\bar{\mathbf{Y}}$ are close enough that all steps of the proof still hold with $\mathbf{Y}$ replaced by $\bar{\mathbf{Y}}$. With that noted, let $\mathbf{y}^1, \ldots, \mathbf{y}^p \in \mathbb{R}^n$ denote the transposes of the rows of $\mathbf{Y}$, and note that these are independent random vectors. From (11), we can see one step of the ADM algorithm takes the form:

$$\mathbf{q}^{(k+1)} = \frac{\frac{1}{p}\sum_{i=1}^{p}\mathbf{y}^i \mathcal{S}_\lambda[(\mathbf{y}^i)^\top \mathbf{q}^{(k)}]}{\left\|\frac{1}{p}\sum_{i=1}^{p}\mathbf{y}^i \mathcal{S}_\lambda[(\mathbf{y}^i)^\top \mathbf{q}^{(k)}]\right\|_2}. \tag{17}$$

This is a very favorable form for analysis: if $\mathbf{q}$ is viewed as fixed, the term in the numerator is a sum of $p$ independent random vectors. To this end, we define a vector valued random process $\mathbf{Q}(\mathbf{q})$ on $\mathbf{q} \in \mathbb{S}^{n-1}$, via

$$\mathbf{Q}(\mathbf{q}) = \frac{1}{p}\sum_{i=1}^{p}\mathbf{y}^i \mathcal{S}_\lambda[(\mathbf{y}^i)^\top \mathbf{q}]. \tag{18}$$

We study the behavior of the iteration (17) through the random process $\mathbf{Q}(\mathbf{q})$. We wish to show that w.h.p. in our choice of $\mathbf{Y}$, $\mathbf{q}^{(k)}$ converges to $(\pm \mathbf{e}_1)$, so that the algorithm successfully retrieves the sparse vector $\mathbf{x}_0 = \mathbf{Y}\mathbf{e}_1$. Thus, we hope that in general, $\mathbf{Q}(\mathbf{q})$ is more concentrated on the first coordinate than $\mathbf{q}$. Let us partition the vector $\mathbf{q}$ as $\mathbf{q} = \begin{bmatrix} q_1 \\ \mathbf{q}_2 \end{bmatrix}$, with $q_1 \in \mathbb{R}$ and $\mathbf{q}_2 \in \mathbb{R}^{n-1}$, and correspondingly partition $\mathbf{Q}(\mathbf{q}) = \begin{bmatrix} Q_1(\mathbf{q}) \\ \mathbf{Q}_2(\mathbf{q}) \end{bmatrix}$, where

$$Q_1(\mathbf{q}) = \frac{1}{p} \sum_{i=1}^{p} x_{0i} S_\lambda \left[ \left( \mathbf{y}^i \right)^\top \mathbf{q} \right] \qquad \text{and} \qquad \mathbf{Q}_2(\mathbf{q}) = \frac{1}{p} \sum_{i=1}^{p} \mathbf{g}^i S_\lambda \left[ \left( \mathbf{y}^i \right)^\top \mathbf{q} \right]. \tag{19}$$

The inner product of $\mathbf{Q}(\mathbf{q})/\|\mathbf{Q}(\mathbf{q})\|_2$ and $\mathbf{e}_1$ is strictly larger than the inner product of $\mathbf{q}$ and $\mathbf{e}_1$ if and only if

$$\frac{|Q_1(\mathbf{q})|}{|q_1|} > \frac{\|\mathbf{Q}_2(\mathbf{q})\|_2}{\|\mathbf{q}_2\|_2}. \tag{20}$$

In the appendix, we show that with high probability, this inequality holds uniformly over a significant portion of the sphere, so the algorithm moves in the correct direction. To complete the proof of Theorem 4.1, we combine the following observations:

**1. Algorithm 1 converges**.

**2. Rounding succeeds when $|r_1| > 2\sqrt{\theta}$.** With high probability, the linear programming based rounding (13) will produce $\pm x_0$, up to scale, whenever it is provided with an input $\mathbf{r}$ whose first coordinate has magnitude at least $2\sqrt{\theta}$.

**3. No jumps away from the caps**. With high probability, for all $\mathbf{q}$ such that $|q|_1 > C_\star \sqrt{\theta}$,

$$\frac{|Q_1(\mathbf{q})|}{\sqrt{|Q_1(\mathbf{q})^2| + \|\mathbf{Q}_2(\mathbf{q})\|_2^2}} \geq 2\sqrt{\theta}. \tag{21}$$

**4. Uniform progress away from the equator**. With high probability, for every $\mathbf{q}$ such that $\frac{1}{2\sqrt{\theta n}} \leq |q_1| \leq C_\star \sqrt{\theta}$, the bound

$$\frac{|Q_1(\mathbf{q})|}{|q_1|} - \frac{\|\mathbf{Q}_2(\mathbf{q})\|_2}{\|\mathbf{q}\|_2} > \frac{c}{np} \tag{22}$$

holds. This implies that if at any iteration $k$ of the algorithm, $|q_1^{(k)}| > \frac{1}{2\sqrt{\theta n}}$, the algorithm will eventually obtain a point $\mathbf{q}^{(k')}$, $k' > k$, for which $|q_1^{(k')}| > C_\star \sqrt{\theta}$.[4]

**5. Location of stationary points**. Steps 1, 3 and 4 above imply that if Algorithm 1 ever obtains a point $\mathbf{q}^{(k)}$ with $|q_1^{(k)}| > \frac{1}{2\sqrt{\theta n}}$, it will converge to a point $\bar{q}$ with $\bar{q}_1 > C_\star \sqrt{\theta}$, provided $\frac{1}{2\sqrt{\theta n}} < 2\sqrt{\theta}$ (i.e., $\theta > \frac{1}{4\sqrt{n}}$).

**6. Good initializers**. With high probability, at least one of the initializers $\mathbf{q}^{(0)}$ satisfies $|q_1^{(0)}| > \frac{1}{2\sqrt{\theta n}}$.

Taken together, these claims imply that from at least one of the initializers $\mathbf{q}^{(0)}$, the ADM algorithm will produce an output $\bar{\mathbf{q}}$ which is accurate enough for LP rounding to exactly return $\mathbf{x}_0$, up to scale. As $\mathbf{x}_0$ is the sparsest nonzero vector in the subspace $\mathcal{S}$ with overwhelming probability, it will be selected as $\mathbf{Y}\mathbf{q}_{i^\star}$, and hence produced by the algorithm.

## 5  Experimental Results

In this section, we show the performance of the proposed ADM algorithm on both synthetic and real datasets. On the synthetic dataset, we show the phase transition of our algorithm on both the planted sparse vector and dictionary learning models; for the real dataset, we demonstrate how seeking sparse vectors can help discover interesting patterns.

## 5.1 Phase Transition on Synthetic Data

For the planted sparse model, for each pair of $(k, p)$, we generate the $n$ dimensional subspace $\mathcal{S} \in \mathbb{R}^p$ by a $k$ sparse vector $\mathbf{x}_0$ with nonzero entries equal to $1$ and a random Gaussian matrix $\mathbf{G} \in \mathbb{R}^{p \times (n-1)}$ with $G_{ij} \stackrel{i.i.d.}{\sim} \mathcal{N}(0, 1/p)$, so that the basis $\mathbf{Y}$ of the subspace $\mathcal{S}$ can be constructed by $\mathbf{Y} = GS([\mathbf{x}_0, \mathbf{G}])\, \mathbf{U}$, where $GS(\cdot)$ denotes the Gram-Schmidt orthonormalization operator and $\mathbf{U} \in \mathbb{R}^{n \times n}$ is an arbitrary orthogonal matrix. We fix the relationship between $n$ and $p$ as $p = 5n \log n$, and set the regularization parameter in (8) as $\lambda = 1/\sqrt{p}$. We use all the normalized rows of $\mathbf{Y}$ as initializations of $\mathbf{q}$ for the proposed ADM algorithm, and run every program for 5000 iterations. We assume the proposed method to be success whenever $\left\| \frac{\mathbf{x}_0}{\|\mathbf{x}_0\|_2} - \mathbf{Yq} \right\|_2 \le \epsilon$ for at least one of the $p$ programs, for some error tolerance $\epsilon = 10^{-3}$. For each pair of $(k, p)$, we repeat the simulation for 5 times.

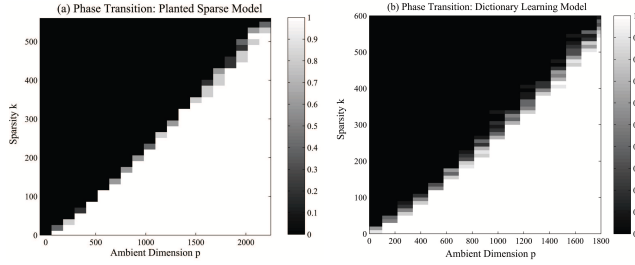

**Figure 1:** Phase transition for the planted sparse model (left) and dictionary learning (right) using the ADM algorithm, with fixed relationship between $p$ and $n$: $p = 5n \log n$. White indicates success and black indicates failure.

Second, we consider the same dictionary learning model as in [24]. Specifically, the observation is assumed to be $\mathbf{Y} = \mathbf{A_0 X_0}$ where $\mathbf{A_0}$ is a square, invertible matrix, and $\mathbf{X_0}$ a $n \times p$ sparse matrix. Since $\mathbf{A_0}$ is invertible, the row space of $\mathbf{Y}$ is the same as that of $\mathbf{X_0}$. For each pair of $(k, n)$, we generate $\mathbf{X_0} = [\mathbf{x}_1, \cdots, \mathbf{x}_n]^\top$, where each vector $\mathbf{x}_i \in \mathbb{R}^p$ is $k$-sparse with every nonzero entry following i.i.d. Gaussian distribution, and construct the observation by $\mathbf{Y}^\top = GS(\mathbf{X}_0^\top)\, \mathbf{U}^\top$. We repeat the same experiment as for the planted sparse model presented above. The only difference is that we assume the proposed method to be success as long as one sparse row of $\mathbf{X}_0$ is recovered by those $p$ programs.

Fig. 1 shows the phase transition between the sparsity level $k = \theta p$ and $p$ for both models. It seems clear for both problems our algorithm can work well into (beyond) the linear regime in sparsity level. Hence for the planted sparse model, to close the gap between our algorithm and practice is one future direction. Also, how to extend our analysis for dictionary learning is another interesting direction.

## 5.2 Exploratory Experiments on Faces

It is well known in computer vision convex objects only subject to illumination changes produce image collection that can be well approximated by low-dimensional space in raw-pixel space [9]. We will play with face subspaces here. First, we extract face images of one person (65 images) under different illumination conditions. Then we apply *robust principal component analysis* [12] to the data and get a low dimensional subspace of dimension 10, i.e., the basis $\mathbf{Y} \in \mathbb{R}^{32256 \times 10}$. We apply the ADM algorithm to find the sparsest element in such a subspace, by randomly selecting $10\%$ rows as initializations for $\mathbf{q}$. We judge the sparsity in a $\ell^1/\ell^2$ sense, that is, the sparsest vector $\hat{\mathbf{x}}_0 = \mathbf{Yq}^*$ should produce the smallest $\|\mathbf{Yq}\|_1 / \|\mathbf{Yq}\|_2$ among all results. Once some sparse vectors are found, we project the subspace onto orthogonal complement of the sparse vectors already found, and continue the seeking process in the projected subspace. Fig. 2 shows the first four sparse vectors we get from the data. We can see they correspond well to different extreme illumination conditions.

Second, we manually select ten different persons' faces under the normal lighting condition. Again, the dimension of the subspace is 10 and $\mathbf{Y} \in \mathbb{R}^{32256 \times 10}$. We repeat the same experiment as stated above. Fig. 3 shows four sparse vectors we get from the data. Interestingly, the sparse vectors roughly

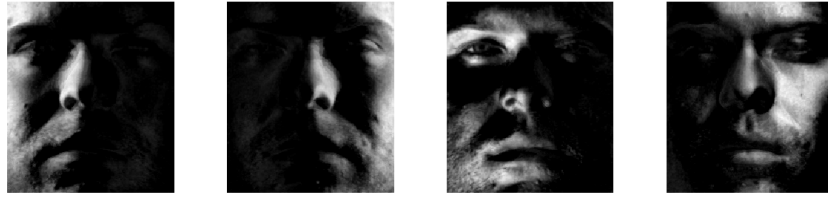

**Figure 2:** Four sparse vectors extracted by the ADM algorithm for one person in the Yale B database under different illuminations.

correspond to differences of face images concentrated around facial parts that different people tend to differ from each other.

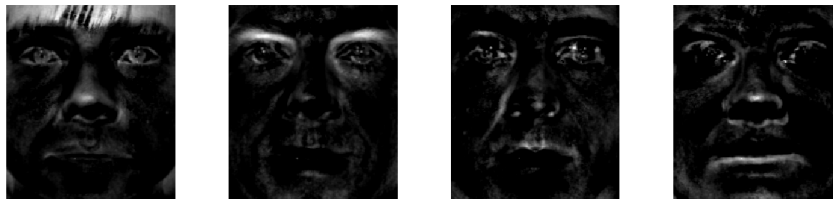

**Figure 3:** Four sparse vectors extracted by the ADM algorithm for 10 persons in the Yale B database under normal illuminations.

In sum, our algorithm seems to find useful sparse vectors for potential applications, like peculiar discovery in first setting, and locating differences in second setting. Netherless, the main goal of this experiment is to invite readers to think about similar pattern discovery problems that might be cast as searching for a sparse vector in a subspace. The experiment also demonstrates in a concrete way the practicality of our algorithm, both in handling data sets of realistic size and in producing attractive results even outside of the (idealized) planted sparse model that we adopt for analysis.

## 6 Discussion

The random models we assume for the subspace can be easily extended to other random models, particularly for dictionary learning. Moreover we believe the algorithm paradigm works far beyond the idealized models, as our preliminary experiments on face data have clearly shown. For the particular planted sparse model, the performance gap in terms of $(p, n, \theta)$ between the empirical simulation and our result is likely due to analysis itself. Advanced techniques to bound the empirical process, such as decoupling [17] techniques, can be deployed in place of our crude union bound to cover all iterates. Our algorithmic paradigm as a whole sits well in the recent surge of research endeavors in provable and practical nonconvex approaches towards many problems of interest, often in large-scale setting [13, 22, 20, 23, 26]. We believe this line of research will become increasingly important in theory and practice. On the application side, the potential of seeking sparse/structured element in a subspace seems largely unexplored, despite the cases we mentioned at the start. We hope this work can invite more application ideas.

## Footnotes

[1]This breakdown behavior is again in sharp contrast to the standard sparse approximation problem (with $\mathbf{b} \neq \mathbf{0}$), in which it is possible to handle very large fractions of nonzeros (say, $\theta = \Omega(1/\log n)$, or even $\theta = \Omega(1)$) using a very simple $\ell^1$ relaxation [14, 18]

[2]More precisely, in our models, random initialization *does* work, but only when the subspace dimension $n$ is extremely low compared to the ambient dimension $p$.

[3]This is the common heuristic that "tall random matrices are well conditioned" [25].

[4]In fact, the rate of progress guaranteed in (22) can be used to bound the complexity of the algorithm; we do not dwell on this here.

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
