[Reviews · NeurIPS 2014]

Submitted by Assigned_Reviewer_19

The aim of the paper is to produce an algorithm and supporting analysis for the problem of finding a planted sparse vector in a random subspace. The paper motivates this problem as being a close relative of sparse recovery as well as having potential extensions to the dictionary learning model (in which more than one sparse vector is sought). Previous convex approaches to this problem undergo a phase transition when the number of nonzero elements in the sparse vector greatly exceed n^(-1/2). The paper finds an alternating directions based algorithm featuring a data-based initialization and an additional rounding step based on linear programming. This algorithm, in contrast to previous convex-based ones, succeeds (whp) with the target vector having a constant fraction of nonzero entries. The theoretical guarantee requires that the dimension p >= n^4 log^2 n, though empirically it seems to succeed for p linear in n. The paper produces experiments which demonstrate this linear phase transition on synthetic data, as well as showing a novel and "strange" experiment on real data demonstrating its potential role in discovering "unusual vectors" in a subspace.

The paper is argued very clearly for the most part and is of high quality on its own terms. While the theoretical significance of this work is clear, the practical significance is not sufficiently clear. It is unclear to me whether the experiments on real data are of interest to practitioners. Furthermore, although the connection to dictionary learning is alluded to as a possible expansion for future works, this is not well argued for either empirically or theoretically. However, the authors have explained this connection in the rebuttal. Stronger arguments for practical importance will benefit the paper (though the authors have alluded to relations with other problems).

I am not a direct expert in this particular area so I may not be able to fully appreciate its potential impact on various communities. I still find the paper very original, of high theoretical quality and even somewhat intriguing and I thus assigned the first (of two) impact scores. This was clearly the strongest paper I reviewed (among five).

There are few typos that can be corrected:
* "with with" in statement of Theorem 2.1 (it will also be good to define "with overwhelming probability")
* Missing section reference (line 122)
*(8) is not *the* Huber m-estimator over Yq (lines 137-8 are confusing).
* Revise writing after first sentence of 192 (e.g., next sentence is not grammatically complete)

I have read the authors' rebuttal and made very minor changes to my review. I congratulate the authors on their nice paper.
Summary: An interesting and strong theoretical paper on finding a planted sparse vector in a random subspace with a possible connection to the dictionary learning model. In order to appeal to a large audience, it will be beneficial to establish stronger arguments for the practical importance of the paper.

Submitted by Assigned_Reviewer_26

The paper provides theoretical guarantees for recovering the sparsest vector in a known, but randomly generated, subspace.
Specifically, it shows that the sparsest vector can be recovered even if the sparsity level is proportional to the dimensional of the vector (an important result) provided the dimensionality of the space is very high (the fourth power) compared to the dimensionality of the subspace.
This recovery is shown to be possible by a simple alternating minimization algorithm with a carefully developed initialization strategy.

Pros.
- Very well written paper. The exposition is crisp!

Cons
- Lack of clear applications
- Motivation for Section 5.2 seems rather unclear

1) Might be nice to explicitly state what “n” is ?
For example, the first line in the intro can be rewritten as
“Suppose we are given a n-dimensional linear subspace of …”

2) It seems that prior results have no constraints on the subspace dimension “n” and the signal dim “p”.
In contrast, the results in the paper require p >= n^4 log^2 n.
In some ways, the paper obtains a stronger result (linear scaling of the sparsity) but when the subspace dimension is very small (or equivalently it is restricted to a small part of the ambient space).
It would be nice for the paper to discuss the implications of this.

3) It is not clear what the message of Section 5.2 is. The results and the explanations seem very much out of place.

Line 46: Spielman et. al. to et al.
line 67. Barak et. al. ——> et al.
Line 122. [??]
Line 207. Missing reference
Summary: Well written paper with a strong result.

Submitted by Assigned_Reviewer_41

This work proposes an alternating optimization algorithm to recover the sparest vector in a given subspace. Suppose the subspace satisfies the planted sparse model, the authors proved that with proper initializations, the alternating procedure can recover the sparse vector x_0 with overwhelming probability even when the fraction of nonzero entries scales as $Omega(1)$ provided that p > n^4 log^2(n). Although the theoretical guarantee is a weaker than the results by BKS13, the algorithm proposed in this paper is very practical and can be applied to large scale datasets.

This paper is technically very sound and claims are well supported by theoretical analysis. The paper is clearly written and well-organized. Especially, the sketch of analysis in Section 4.2 conveys a lot of intuitions behind the technical proof to the readers. Although the theory only works for the planted sparse model, I believe this work will be an important one in the field of compressed sensing.

As a future work, it will be very interesting to see more general theoretical results without assuming the planted sparse models. It will be also interesting to see how to extend the current analysis to dictionary learning.

In experimental results in Section 5.1, it is better to provide more details of the data generation process for the dictionary learning setting. E.g., why choose p=5nlog(n) instead of 5n ? Is the success defined by successfully recover the sparsity pattern for all sparse vectors ? What is the size of the dictionary ?

Another minor comment: in Line 314, it should be performance instead of performane.
Summary: This paper proposed a practical algorithm to recover the sparsest vector in a subspace with pretty strong theoretical results. Although the proof is rather technical, the presentation of the sketch of analysis is great and successfully conveys intuitions behind the proof.
Author Feedback
Author rebuttal: We would like to express our gratitude to the anonymous reviewers for your careful review, insightful comments, and strong recommendations of our paper. We address the major concerns raised by the reviewers as follows.

* Interpretation of Our Results: We want to take the chance to clarify the following points.

- For the planted sparse model, all results we know put some constraints on the relationship between n and p. In particular, the best results currently known [BKS13] show that a certain sum-of-squares relaxation, rounded in a novel way, can recover the sparse vector x_0 when p > Cn^2 and \theta = \Omega(1).

- For the L2 constrained version in Eq (5), our analysis on the global optimality condition shows that x_0 and -x_0 are the only global minimizers w.h.p. when p>Cn and \theta=\Omega(1) (Theorem 2.1).

- The proposed ADM algorithm solves a relaxed version of Eq. (5), i.e., Eq. (8). Our analysis shows solving (8) with \lambda = 1/\sqrt{p} using the ADM algorithm plus rounding recovers the sparsest vector x_0 w.h.p. when p > Cn^4 \log^2(n) and \theta = \Omega(1) (Theorem 4.1).

- There is an obvious gap between the scaling of p wrt n under which our results guarantee global optimality of the target solution and the scaling under which our results guarantee correct recovery by the ADM algorithm. Our experiment in Sec 5.1 suggests that our algorithm succeeds in recovery even when p ~ Cn \log n. So the gap is most likely due to limitations of our analysis, rather than fundamental limitations of our algorithm. Tightening the dependency is left for our future work.

- The complexity of the SDP approach of [BKS13] is a high-degree polynomial in p and n, seemingly precluding most practical applications of interest to the NIPS community. Although our theoretical guarantee is weaker than [BKS13], the ADM algorithm we propose is very practical and can be applied to large-scale datasets.

* Applications and Real Data Experiment: In the introduction, we briefly allude to the fact that the problem of finding the sparsest vector in a linear subspace arises in numerical linear algebra, latent variable identification, and blind source separation. We believe that the practical algorithm and theoretical guarantees introduced here will stimulate additional application ideas, both within these domains and beyond.

One of the main goals of the real data experiment is to invite the reader to think about similar pattern discovery problems that might be cast as searching for a sparse vector in a subspace. The experiment also demonstrates in a concrete way the practicality of our algorithm, both in handling data sets of realistic size and in producing attractive results even outside of the (idealized) planted sparse model that we adopt for analysis.

* Details of Experiment Setup for Dictionary Learning (DL) (Sec 5.1): This experiment is exploratory in nature and we have omitted some details. The data model is the same as that assumed in [SWW12]. Specifically, the observation is assumed to be Y = A_0 X_0, where A_0 is assumed to be a square, invertible matrix and X_0 a n x p matrix its entries obeying the i.i.d. Bernoulli-Gaussian model with parameter \theta. With high probability, all n rows of X_0 are sparse. Since A_0 is invertible, the row space of Y is the same as that of X_0. Hence we hope to recover (up to scale and sign ambiguity) some rows of X_0 by seeking the sparsest vector from the subspace formed by rows of Y. We consider a successful recovery as long as one of the sparse rows of X_0 get recovered, in the sense the vector direction (or its reverse) is correctly identified. We set p = C n*log n because this is the minimal p required for any algorithm to recover any sparse vector under the i.i.d Bernoulli-Gaussian model for DL, as argued in [SWW12]. We will add additional details on this experiment in the final version.

* Extension to Dictionary Learning (DL): One straightforward way to extend the algorithm to DL is to extract the n sparse rows of X_0 sequentially. This can be (optionally) combined with a “deflation” step that ensures that the sparse vectors obtained are linearly independent. To prove that the algorithm succeeds in the Bernoulli-Gaussian model, one can follow a similar pattern of argument to that used here – namely, to argue that the algorithm’s behavior is predicted by a certain large sample limit. However, the argument is more delicate, due to the presence of many (2n) desirable solutions. Inspired by the results in this NIPS submission, we are currently working on an analysis that covers both of these cases, as well as other nonconvex problems of a similar nature.

* Typos: We thank the reviewers for pointing out several typos, which we will carefully correct in the final version.